# UPLC-MS/MS of Atractylenolide I, Atractylenolide II, Atractylenolide III, and Atractyloside A in Rat Plasma after Oral Administration of Raw and Wheat Bran-Processed Atractylodis Rhizoma

**DOI:** 10.3390/molecules23123234

**Published:** 2018-12-07

**Authors:** Shizhao Xu, Xiaojie Qi, Yuqiang Liu, Yuhan Liu, Xin Lv, Jianzhi Sun, Qian Cai

**Affiliations:** College of Pharmacy, Liaoning University of Traditional Chinese Medicine, Dalian 116000, China; dazhao666@163.com (S.X.); qixiaojiemail@163.com (X.Q.); liuyuqiang@126.com (Y.L.); lyhown@163.com (Y.L.); lvxiaoxin1222@163.com (X.L.); 15804268722@163.com (J.S.)

**Keywords:** Atractylodis Rhizoma, wheat bran processing, integral pharmacokinetics, UPLC-MS/MS

## Abstract

Atractylodis Rhizoma is the dried rhizome of *Atractylodes lancea* (Thunb.) DC. or *Atractylodes chinensis* (DC.) Koidz and is often processed by stir-frying with wheat bran to reduce its dryness and increase its spleen tonifying activity. However, the mechanism by which the processing has this effect remains unknown. To explain the mechanism based on the pharmacokinetics of the active compounds, a rapid, sensitive ultra-performance liquid chromatography-tandem mass spectrometry method was developed to analyze atractylenolides I, II, and III, and atractyloside A simultaneously in rat plasma after oral administration of raw and processed Atractylodis Rhizoma. Acetaminophen was used as the internal standard and the plasma samples were pretreated with methanol. Positive ionization mode coupled with multiple reaction monitoring mode was used to analyze the four compounds. The method validation revealed that all the calibration curves displayed good linear regression over the concentration ranges of 3.2–350, 4–500, 4–500, and 3.44–430 ng/mL for atractylenolides I, II, and III, and atractyloside A, respectively. The relative standard deviations of the intra- and inter-day precisions of the four compounds were less than 6% with accuracies (relative error) below 2.38%, and the extraction recoveries were more than 71.90 ± 4.97%. The main pharmacokinetic parameters of the four compounds were estimated with Drug and Statistics 3.0 and the integral pharmacokinetics were determined based on an area under the curve weighting method. The results showed that the integral maximum plasma concentration and area under the curve increased after oral administration of processed Atractylodis Rhizoma.

## 1. Introduction

The dried rhizome of *Atractylodes lancea* (Thunb.) DC or *A. chinensis* (DC.) Koidz has been widely used as an herbal medicine for thousands of years [1]. Atractylenolides I, II, and III, and atractyloside A, which are marker compounds in *A. lancea*, have attracted interest in China and worldwide owing to their anti-inflammatory [2,3,4] and anticancer activities [5], and their ability to increase gastrointestinal peristalsis [6]. Wang et al. found that atractylenolide I may possess anticancer activity by inducing apoptosis via Cu, Zn-superoxide dismutase inhibition in HL-60 cells [7]. In addition, Dong et al. proposed that atractylenolides I and III have anti-inflammatory activity based on their inhibition of ear edema induced by xylene in mice [8]. Chen et al. reported that atractylenolide I can be used to treat rhubarb-induced spleen deficiency in rats [9].

In a previous study, we found that processing the rhizome decreased the active compound content and increased the spleen tonifying activity. Xue et al. [10] suggested that processing with wheat bran could promote the therapeutic effect on spleen deficiency. Chang et al. [11] performed a pharmacokinetic study of atractylodin after oral administration of raw and wheat bran-processed Atractylodis Rhizoma and demonstrated that stir-frying the rhizome with wheat bran promoted and accelerated the absorption of atractylodin.

Many analytical methods have been used for the simultaneous determination of one or several bioactive compounds, such as atractylenolides I, II, and III, in rat plasma or tissue after oral administration of raw and wheat bran-processed Atractylodis Rhizoma [12]. For instance, Chen et al. [13,14] used a sensitive high-performance liquid chromatography (HPLC) approach to study the pharmacokinetics and tissue distribution characteristics of atractylenolide I in rats. Zhu et al. [15] simultaneously measured atractylenolides I, II, and III and their pharmacokinetics in rats by HPLC-mass spectrometry. However, these HPLC pharmacokinetic studies examined atractylenolides I, II, and III in Atractylodis Macrocephalae Rhizoma, whereas little is known about the pharmacokinetics of atractylenolides I, II, and III, and atractyloside A in Atractylodis Rhizoma. Furthermore, these studies focused on the pharmacokinetics of a single compound, which does not reflect the synergistic effects common in traditional Chinese medicine. Many studies of the mechanism of wheat bran processing have shown that wheat bran processing decreases the content of atractylenolides I, II, and III, and atractyloside A [16,17] and increases the spleen tonifying effect. Therefore, it is important to establish a method for determining multiple compounds in rat plasma after oral administration of raw and wheat bran-processed Atractylodis Rhizoma and use the method to study the integral pharmacokinetics of these compounds in rat plasma.

In this study, we established an ultra-performance liquid chromatography-tandem mass spectrometry (UPLC-MS/MS) method to determine the content of atractylenolides I, II, and III, and atractyloside A in plasma after rats were orally administered raw or wheat bran-processed Atractylodis Rhizoma. We compared the pharmacokinetic parameters by using an integral pharmacokinetic method. This research provides useful information about the processing mechanism of Atractylodis Rhizoma.

## 2. Results

### 2.1. Method Validation

#### 2.1.1. Selectivity

Typical chromatograms obtained from blank plasma, blank plasma spiked with four standard analytes and the IS, and plasma samples obtained 1.5 h after oral administration of raw and wheat bran-processed Atractylodis Rhizoma are shown in Figure 1, Figure 2, Figure 3, Figure 4 and Figure 5, demonstrating that there was no major interference from endogenous compounds in the analysis of the compounds, and good selectivity was achieved. The retention times of atractylenolides I, II, and III, atractyloside A, and the IS were approximately 7.16, 7.58, 6.83, 3.16, and 2.62 min, respectively. The total run time was 10 min.

#### 2.1.2. Linearity and Sensitivity

All the linear regressions of atractylenolides I, II, and III, and atractyloside A in rat plasma displayed good linear relationships over ranges of 3.2–350, 4–500, 4–500, and 3.44–430 ng/mL, respectively. The slope and intercept of calibration graphs were calculated by weighted (1/*x*) least-squares linear regression. The lower LOQs, defined as the lowest concentration on the calibration curves and determined at a signal-to-noise ratio of >10, were 3.2 ng/mL for atractylenolide I, 4 ng/mL for atractylenolide II, 4 ng/mL for atractylenolide III, and 3.44 ng/mL for atractyloside A, with both precision and accuracy not exceeding 15% (Table 1).

#### 2.1.3. Precision and Accuracy

The intra- and inter-day precisions were satisfactory, with RSD values of less than 7.31 and 9.2%, respectively, and RE values from 2.75 to −5.81% (Table 2).

#### 2.1.4. Extraction Recovery and Matrix Effect

The extraction recoveries of atractylenolides I, II, and III, and atractyloside A were no less than 71.90 ± 4.97% (Table 3), and that of the IS was 98.54 ± 6.81%, suggesting that the precision and accuracy of this method were acceptable. The matrix effects of the four compounds at the three QC levels ranged from 69.25 to 120.28%, which indicated that the matrix effect was negligible.

#### 2.1.5. Stability

The results for the short-term stability and freeze-thaw stability are shown in Table 4 and Table 5. There was no major degradation of the five compounds under the different storage conditions.

### 2.2. Integrated Pharmacokinetics Study

The validated UPLC-MS/MS method was used to measure atractylenolides I, II, and III, and atractyloside A simultaneously in rat plasma after oral administration of raw and wheat bran-processed Atractylodis Rhizoma at a dose of 3.75 g/kg. The integrated pharmacokinetics data were estimated by Drug and Statistics 3.0 (BioGuider Medicinal Technology Co. Ltd, Shanghai, China). The AUC_0–∞_ values and corresponding weight coefficients (*W*_j_) are shown in Table 6. The mean and integrated plasma concentration-time profiles of raw and wheat bran-processed Atractylodis Rhizoma for atractylenolides I, II, and III, and atractyloside A are shown in Figure 6. The corresponding pharmacokinetic parameters are summarized in Table 7.

## 3. Discussion

The mean *C*_max_ values of atractylenolides I, II, and III, and atractyloside A (66.94 ± 10.89, 55.9 ± 13.58, 113.10 ± 19.04, and 69.38 ± 8.29 ng/mL, respectively) in the wheat bran-processed Atractylodis Rhizoma group were 2.09-, 1.13-, 1.30-, and 1.20-fold greater than those in the raw Atractylodis Rhizoma group (Table 6). The AUC_0–t_ values of atractylenolides I, II, and III, and atractyloside A (219.14 ± 46.65, 202.43 ± 68.52, 284.83 ± 32.94, and 306.91 ± 73.75 h μg/L) were 1.88-, 1.12-, 1.24-, and 2.17-fold greater than those in the raw Atractylodis Rhizoma group. As expected, the significant increase in the *C*_max_ and AUC_0–t_ values of the four compounds in the plasma of wheat bran-processed Atractylodis Rhizoma group confirmed that wheat bran processing increases the plasma exposures of these compounds, which agrees with our previous studies [18]. Wheat bran processing increases the spleen tonifying activity of Atractylodis Rhizoma [10]. The significant increase in the plasma exposure levels of these four compounds in the wheat bran-processed group may have caused the substantial increase in the spleen tonifying activity.

Multiple compounds in traditional Chinese medicine often show synergistic effects. Based on the idea of integrated pharmacokinetics, we monitored the pharmacokinetic characteristics of atractylenolides I, II, and III, and atractyloside A in rat plasma after oral administration of raw and wheat bran-processed Atractylodis Rhizoma, to obtain a complete picture of the pharmacokinetic characteristics of the compounds. The integral *C*_max_ (70.02 ± 5.16 ng/mL) in the wheat bran-processed Atractylodis Rhizoma group was 1.50-fold greater than that in the raw Atractylodis Rhizoma group, and the integral AUC_0–t_ (263.23 ± 40.15 h·μg/L) in the wheat bran-processed Atractylodis Rhizoma group was 1.29-fold greater than that in the raw Atractylodis Rhizoma group. The results confirmed that processing Atractylodis Rhizoma with wheat bran promotes and accelerates the absorption of compounds compared with the raw Atractylodis Rhizoma.

The pharmacokinetic subjects in present study were normal rats, but the metabolic processes of Atractylodis Rhizoma in diseased rats may be different. So, if the pharmacokinetics of spleen deficiency model rats is studied, the results may be more meaningful. In addition, the composition of traditional Chinese medicines is complex, so there may be limitations in studying only four of these and more ingredients should be studied in the future.

## 4. Materials and Methods

### 4.1. Chemicals and Reagents

Atractylodis Rhizoma was collected from Luotian County (Huanggang, Hubei Province, China) and identified as the dried rhizome of *A. lancea* Thunb. DC. by Professor Feng Li (TCM Identification Department, Liaoning University of Traditional Chinese Medicine, Dalian, China). The method of stir-frying with wheat bran was based on the Chinese Pharmacopoeia 2015. Atractylenolides I, II, and III, atractyloside A, and acetaminophen were obtained from Yongjian Pharmaceutical Technology Co., Ltd. (Taizhou, Jiangsu, China; purities >98%). The chemical structures of these compounds are shown in Figure 7A–E. HPLC grade formic acid and acetonitrile were purchased from Thermo Fisher Scientific (Waltham, MA, USA). Ultrapure water from a Milli-Q system (Merck-Millipore, Burlington, MA, USA) was used. All other chemicals were of analytical reagent grade and were purchased from Sinopharm Chemical Reagent Co., Ltd. (Suzhou, Jiangsu, China).

Male Sprague Dawley rats (200 ± 20 g) were obtained from Liaoning Changsheng Biological Technology Co., Ltd. (Benxi, China), and were housed in an environmentally controlled breeding room and fed with standard laboratory food and water *ad libitum* for a week before staring the experiments. The animals were fasted for 12 h before drug administration and water was freely available to the rats during the experiments. Animal experiments were carried out in accordance with the Guidelines for Animal Experimentation of Liaoning University of Traditional Chinese Medicine, and the procedure was approved by the Animal Ethics Committee of this institution (2017-125).

### 4.2. Optimization of UPLC-MS/MS Conditions

A UPLC system (Acquity, Waters, Milford, MA, USA) coupled with a mass spectrometer detector (XEVO-TQ-S, Waters) was used. The separation was performed on an Acquity UPLC BEH C18 column (Waters; 2.1 × 100 mm, 1.7 µm). The mobile phase consisted of acetonitrile (solution A) and ultrapure water with 0.1% methanoic acid (solution B). The gradient elution program was 5–10% B (0–1.5 min), 10–25% B (1.5–3 min), 25–80% B (3–6 min), 80% B (6–8 min), 80–5% B (8–8.5 min), and 5% B (8.5–10 min). The mobile phase was delivered at a flow rate of 0.30 mL/min. The column and autosampler temperature were maintained at 40 and 10 °C, respectively. The injection volume was 2 µL and the eluent was detected by mass spectrometry with electrospray ionization (ESI) in positive scan mode. The optimized detector parameters were as follows: positive ESI with a capillary voltage of 3.0 kV; desolvation temperature of 400 °C; desolvation gas flow of 800 L/h, core gas flow of 150 L/h, and nebulizer gas flow of 7 bar. Selected multiple reaction monitoring (MRM), using the precursor to product ion combinations of *m/z* 233.3→187.2, 231.2→185.2, 249.2→231.2, 471.3→203.2, 151.9→110.16, was used to detect atractylenolides I, II, and III, atractyloside A, and the internal standard (IS), acetaminophen, respectively (Table 8). MRM chromatograms of the five compounds are shown in Figure 8.

### 4.3. Wheat Bran Processing Procedure

According to the general rules of processing in the Chinese Pharmacopeia 2015, the wheat bran processing procedure for Atractylodis Rhizoma was performed as follows: raw Atractylodis Rhizoma (100 kg) was stir-fried with wheat bran (10 kg) until the surface of the Atractylodis Rhizoma became dark yellow. The wheat bran-processed Atractylodis Rhizoma was transferred to a cool dry place after the wheat bran was removed.

### 4.4. Preparation of Raw and Wheat Bran-Processed Atractylodis Rhizoma Extracts

Raw or wheat bran-processed Atractylodis Rhizoma (100 g) was powdered and soaked in 95% ethanol (600 mL) for 24 h, and then percolated at 2 mL/min. The ethanol was evaporated to near dryness under reduced pressure to obtain the residue. Distilled water was added to the residue and the solution was vortexed. The final concentration of the Atractylodis Rhizoma solution was 3 g/mL [19]. The sample was stored in a dark dry place before use.

### 4.5. Preparation of Stock Solutions, Calibration Standards, and Quality Control Samples

The stock solutions of atractylenolides I, II, and III, atractyloside A, and the IS were prepared by weighing the reference standards of the five compounds precisely and dissolving them in methanol to yield concentrations of 35, 50, 50, 43, and 52 μg/mL, respectively. A series of standard mixture working solutions with concentrations of 3.2–3500 ng/mL for atractylenolide I, 4–5000 ng/mL for atractylenolide II, 4–5000 ng/mL for atractylenolide III, and 3.44–4300 ng/mL for atractyloside A were obtained by diluting the stock standard solutions with methanol. The IS working solution (52 ng/mL) was prepared by diluting the IS stock solution with methanol. All solutions were stored at 4 °C. Seven calibration solutions of atractylenolide I (3.2, 5.6, 16, 28, 70, 140, 350 ng/mL), atractylenolide II (4, 8, 20, 40, 100, 200, 500 ng/mL), atractylenolide III (4, 8, 20, 40, 100, 200, 500 ng/mL), and atractyloside A (3.44, 6.88, 17.2, 34.4, 86, 172, 430 ng/mL) were prepared by spiking blank rat plasma (200 µL) with the appropriate amount of the standard mixture working solutions (50 µL) and IS working solution (50 µL). Quality control (QC) samples were prepared at low, medium, and high concentrations of 5.6, 28, 140 ng/mL for atractylenolide I; 8, 40, 200 ng/mL for atractylenolide II; 8, 40, 200 ng/mL for atractylenolide III and 6.88, 34.4, 172 ng/mL for atractyloside A, respectively. QC samples were stored at −20 °C.

### 4.6. Plasma Sample Preparation

Plasma samples were extracted by methanol precipitation. The plasma sample (200 µL) was spiked with IS solution (100 µL, 52 ng/mL), extracted with methanol (800 µL) by vortexing for 2 min, centrifuged at 17,108× *g* for 10 min, and then evaporated to dryness under a gentle stream of nitrogen. The residue was reconstituted in the mobile phase (100 µL), vortexed for 1 min, and centrifuged at 17,108× *g* for 10 min. Finally, the supernatant (3 µL) was injected into the UPLC-MS/MS system for analysis.

### 4.7. Method Validation

#### 4.7.1. Selectivity

The selectivity was determined by comparing chromatograms of blank plasma samples obtained from rats with those of the corresponding plasma samples spiked with atractylenolides I, II, and III, atractyloside A, and IS, and plasma samples after oral administration of raw Atractylodis Rhizoma.

#### 4.7.2. Linearity and Sensitivity

The linearities were evaluated over concentration ranges of 3.2–350, 4–500, 4–500, and 3.44–430 ng/mL for atractylenolides I, II, and III, and atractyloside A, respectively. A seven-point standard curve was constructed based on the peak area ratios of atractylenolides I, II, and III, and atractyloside A to the IS versus the concentrations of the compounds. Calibration curves were prepared according to the preparation of calibration standards. The regression equations were obtained by applying weighted least-squares linear regression (1/*x*). The lower limit of quantification (LOQ) was determined by stepwise dilution of the QC samples at the lowest concentration level with a signal-to-noise ratio of ≥10.

#### 4.7.3. Precision and Accuracy

The precision and accuracy of the method were evaluated with QC samples at low, medium, and high concentrations and using four replicates on three consecutive days. The intra-and inter-assay precisions were assessed by determined the QC samples at three concentrations for each compound. For the inter-day validation, four replicates of the QC plasma samples were analyzed on the same day, whereas the inter-day values were determined over three consecutive days. The accepted criteria for each QC sample were that the precision and accuracy should not exceed 15%. The precision was expressed as the relative standard deviation (RSD) and the accuracy as the relative error (RE).

#### 4.7.4. Extraction Recovery and Matrix Effect

The extraction recoveries of atractylenolides I, II, and III, and atractyloside A at three QC sample concentrations were determined using six replicates by comparing the peak area ratios of the compounds to the IS in the post-extraction spiked samples with those acquired from the pre-extraction spiked samples. The matrix effects in the approach were evaluated by comparing the peak areas of the compounds in the post-extraction spiked samples with those of the standard solutions.

#### 4.7.5. Stability

The stabilities of atractylenolides I, II, and III, and atractyloside A in rat plasma were evaluated at three different QC concentrations using six replicates under two conditions, including at 25 °C for 6 h, and three freeze and thaw cycles. Unextracted QC samples of the four compounds at low, medium, and high concentrations were kept at ambient temperature (25 °C) for 6 h to determine the short-term stability of the compounds in rat plasma. The samples were processed and analyzed, and then the concentrations were compared with the nominal values of the QC samples. The stabilities of the plasma samples after three freeze and thaw cycles were determined. In each cycle, the QC samples were stored at −20 °C for 24 h and thawed unassisted at room temperature. When completely thawed, the samples were refrozen within 24 h. The cycle was repeated three times and the samples were analyzed after the third cycle. Stability was assessed by comparing the mean concentration of the stored QC samples with the mean concentration of freshly prepared QC samples.

### 4.8. Animal and Pharmacokinetic Study

Ten rats were randomly divided into two groups (five per group). The two groups were orally administered raw Atractylodis Rhizoma at a dose of 3.75 g/kg (equal to 0.48, 0.6, 0.68, and 3.11 mg/kg of atractylenolides I, II, and III, and atractyloside A, respectively), and wheat bran-processed Atractylodis Rhizoma extract solution (equal to 0.46, 0.75, 0.56, and 3.03 mg/kg of atractylenolides I, II, and III, and atractyloside A, respectively). Plasma samples (0.3 mL) were collected in heparinized tubes from the orbital sinus at 0.083, 0.016, 0.25, 0.5, 1, 1.5,2, 3, 4, 6, 8, 10, and 12 h, and were then immediately centrifuged at 17,108× *g* for 10 min. The supernatant was transferred to clean test tubes and stored at −20 °C until analysis. Pharmacokinetic parameters, including the area under the blood concentration-time curve (*AUC*_0–∞_), elimination half-life (*t*_1/2_), apparent volume of distribution (*V*_d_/*F*), and oral clearance (*CL*/*F*) were calculated by a noncompartmental method using Drug and Statistics 3.0 (Chinese Pharmacological Society, Shanghai, China). Other parameters were obtained directly from the experimental observations.

### 4.9. Pharmacokinetic Parameters and Statistical Analysis

The pharmacokinetic parameters of atractylenolides I, II, and III, and atractyloside A were analyzed by using the one-way ANOVA test with SPSS 17.0 (SPSS Inc., Chicago, IL, USA) and linear regression analysis. The integrated concentrations at each time point for the four compounds were calculated based on the AUC weighting approach using equations (1)–(3). The weighting coefficient (*W*_j_) and the total concentration of each compound (∑*AUC*_0–∞_) were calculated by:
(1)$$w1=\frac{AUC_{0–\infty }}{\sum AUC_{0–\infty }}$$
∑*AUC*_0–∞_ = *AUC*_0–∞_ AT-I + *AUC*_0–∞_ AT-II + *AUC*_0–∞_ AT-III + *AUC*_0–∞_ atractyloside A(2)

The integrated concentration (*C_F_*) for compound *j* was calculated as:*C_F_* = ∑(*W_j_* × *C_j_*)(3)
where *W* is the weighting coefficient and *C* is the plasma concentration at each time point.

## 5. Conclusions

In summary, a UPLS-MS/MS method was established for the simultaneous determination of atractylenolides I, II, and III, and atractyloside A in rat plasma. We used an integral pharmacokinetic approach for evaluating and comparing the pharmacokinetic characteristics of the compounds in plasma after oral administration of raw and wheat bran-processed Atractylodis Rhizoma. The wheat bran processing increased the plasma exposure of these four compounds, which may explain the increase in activity after Atractylodis Rhizoma is processed.

## Figures and Tables

**Figure 1 molecules-23-03234-f001:**
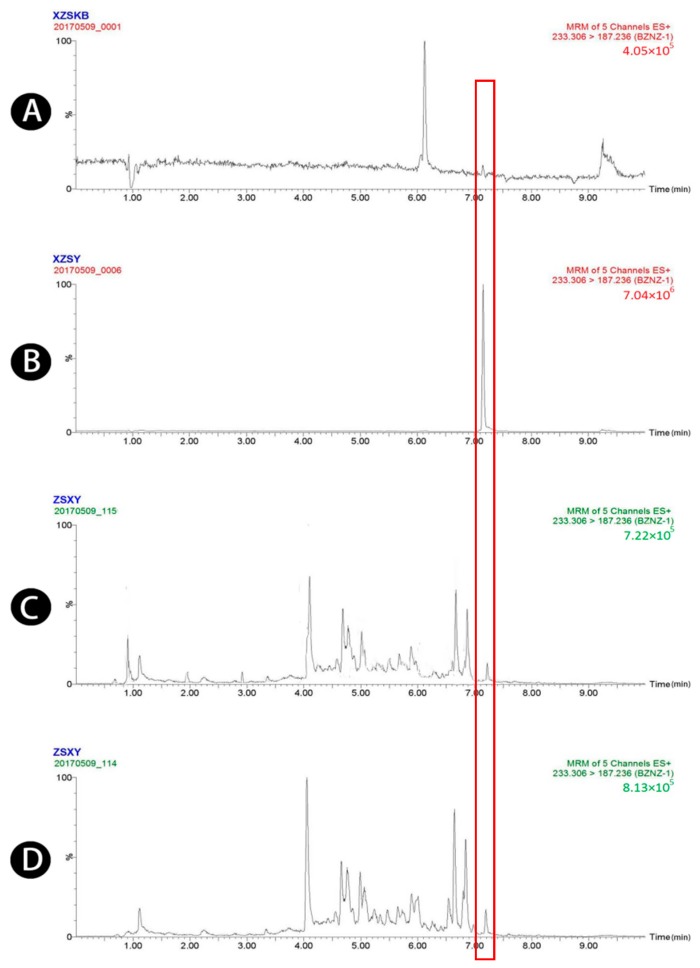
UPLC-MS/MS chromatograms of atractylenolide I (peak 1). (**A**) Blank plasma, (**B**) blank plasma spiked with atractylenolide I at the LOQ of 3.2 ng/mL, (spiked with IS), (**C**) plasma sample 1.5 h after oral administration of raw Atractylodis Rhizoma at a dose of 3.75 g/kg, and (**D**) plasma sample 1.5 h after oral administration of wheat bran-processed Atractylodis Rhizoma at a dose of 3.75 g/kg.

**Figure 2 molecules-23-03234-f002:**
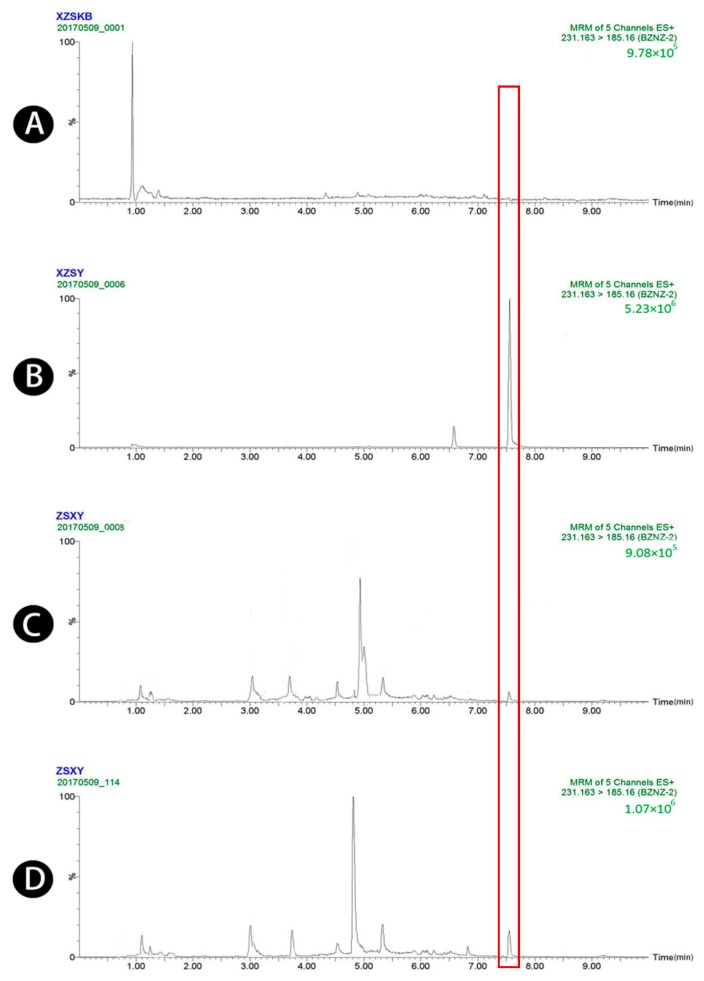
UPLC-MS/MS chromatograms of atractylenolide II (peak 2). (**A**) Blank plasma, (**B**) blank plasma spiked with atractylenolide II at the LOQ of 4 ng/mL (spiked with IS), (**C**) plasma sample 1.5 h after oral administration of raw Atractylodis Rhizoma at a dose of 3.75 g/kg, and (**D**) plasma sample 1.5 h after oral administration of wheat bran-processed Atractylodis Rhizoma at a dose of 3.75 g/kg.

**Figure 3 molecules-23-03234-f003:**
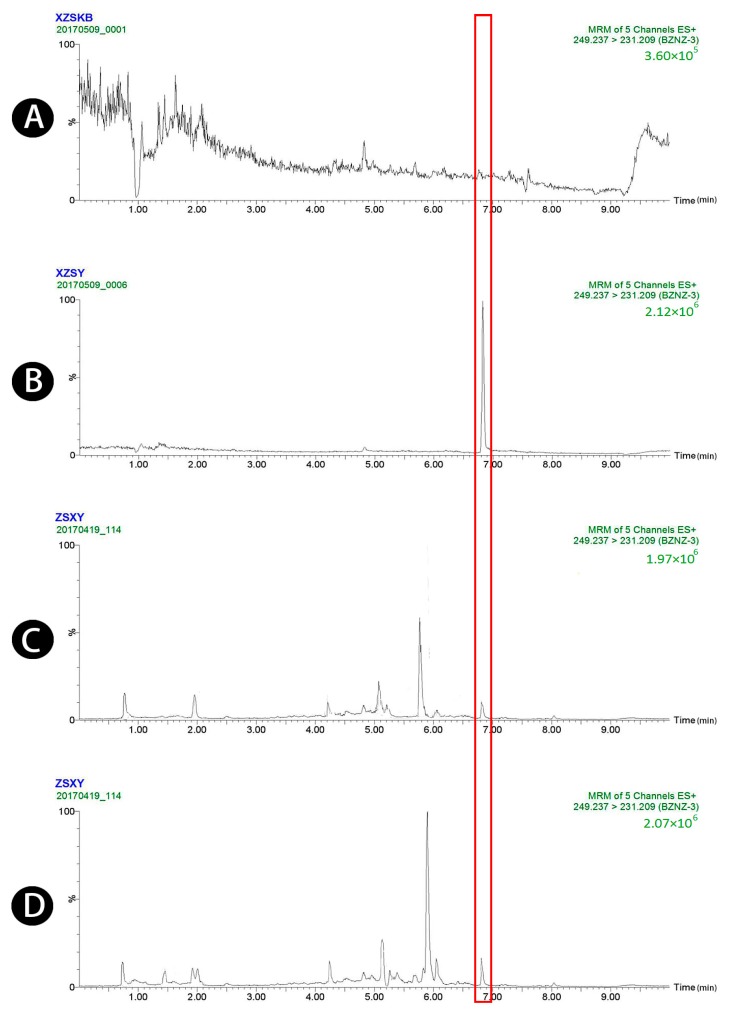
UPLC-MS/MS chromatograms of atractylenolide III (peak 3). (**A**) Blank plasma, (**B**) blank plasma spiked with atractylenolide III at the LOQ of 4 ng/mL (spiked with IS), (**C**) plasma sample 1.5 h after oral administration of raw Atractylodis Rhizoma at a dose of 3.75 g/kg, and (**D**) plasma sample 1.5 h after oral administration of wheat bran-processed Atractylodis Rhizoma at a dose of 3.75 g/kg.

**Figure 4 molecules-23-03234-f004:**
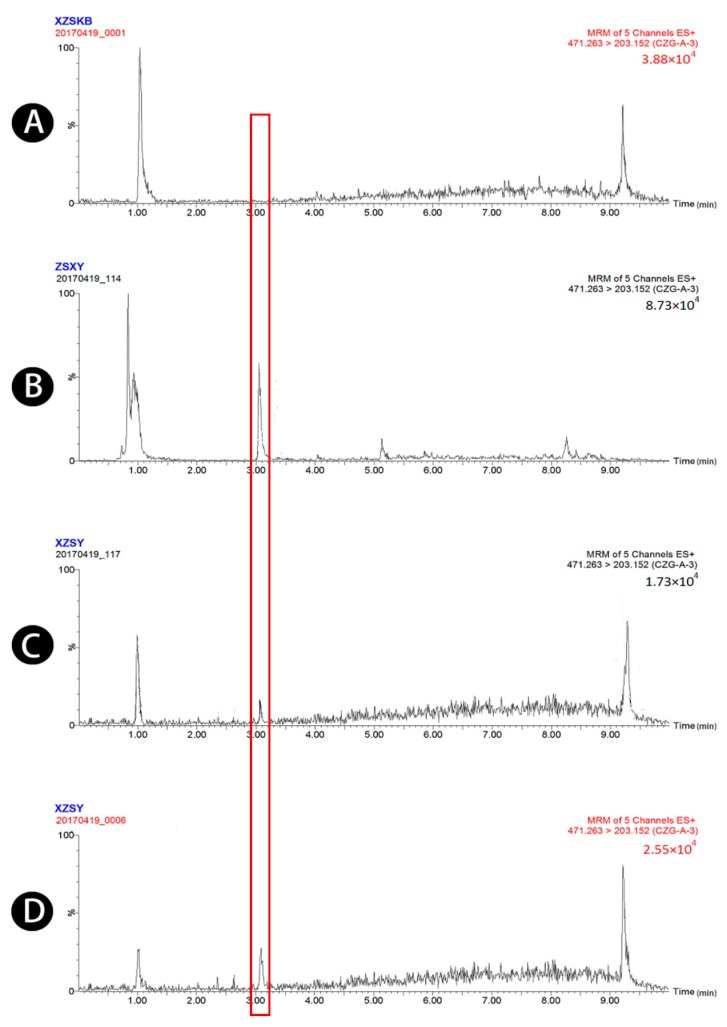
UPLC-MS/MS chromatograms of atractyloside A (peak 4). (**A**) Blank plasma, (**B**) blank plasma spiked with atractyloside A at the LOQ of 3.44 ng/mL (spiked with IS), (**C**) plasma sample 1.5 h after oral administration of raw Atractylodis Rhizoma at a dose of 3.75 g/kg, and (**D**) plasma sample 1.5 h after oral administration of wheat bran-processed Atractylodis Rhizoma at a dose of 3.75 g/kg.

**Figure 5 molecules-23-03234-f005:**
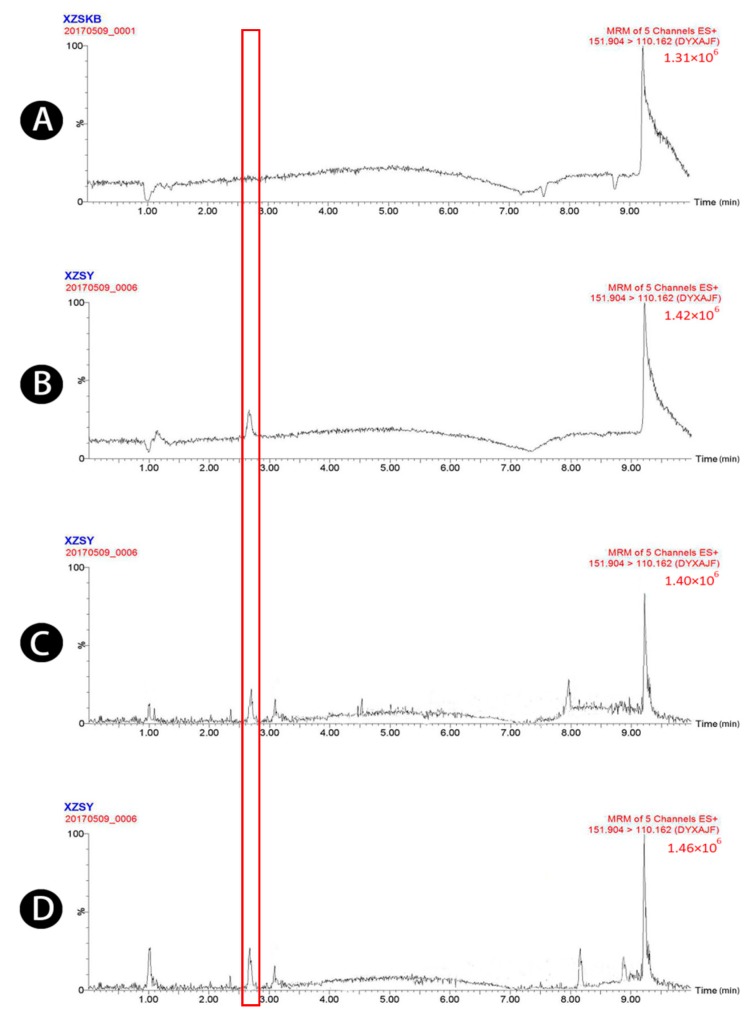
UPLC-MS/MS chromatograms of 4-acetaminophen (peak 5). (**A**) Blank plasma, (**B**) blank plasma spiked with 4-acetaminophen (52 ng/mL), (**C**) plasma sample 1.5 h after oral administration of raw Atractylodis Rhizoma at a dose of 3.75 g/kg, and (**D**) plasma sample 1.5 h after oral administration of wheat bran-processed Atractylodis Rhizoma at a dose of 3.75 g/kg.

**Figure 6 molecules-23-03234-f006:**
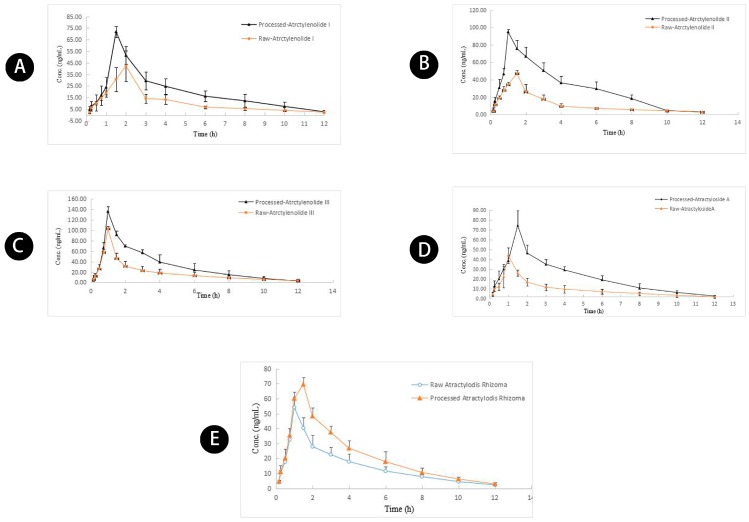
Mean concentration-time profiles of (**A**) atractylenolide I, (**B**) atractylenolide II, (**C**) atractylenolide III, (**D**) atractyloside A, (**E**) AUC-based integrated concentration in rat plasma after oral administration of raw and wheat bran-processed Atractylodis Rhizoma.

**Figure 7 molecules-23-03234-f007:**
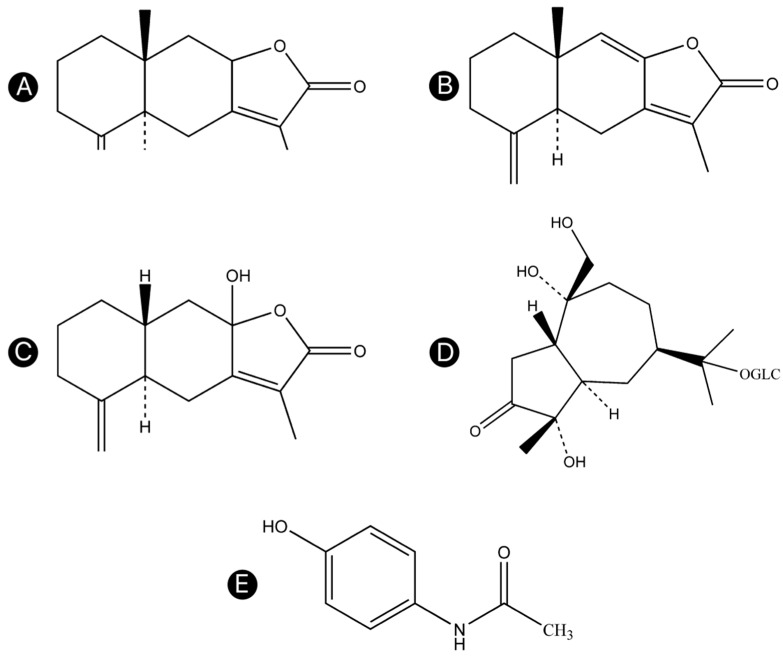
Chemical structures of the four compounds in Atractylodis Rhizoma and the IS. (**A**) Atractylenolide I, (**B**) atractylenolide II, (**C**) atractylenolide III, (**D**) atractyloside A, and (**E**) acetaminophen.

**Figure 8 molecules-23-03234-f008:**
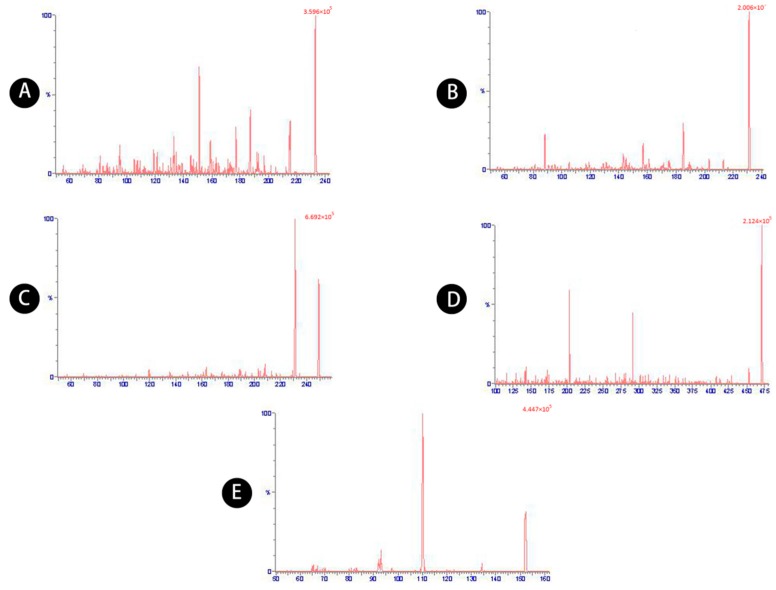
MRM chromatograms of (**A**) atractylenolide I, (**B**) atractylenolide II, (**C**) atractylenolide III, (**D**) atractyloside A, (**E**) acetaminophen.

**Table 1 molecules-23-03234-t001:** Regression, linear range, and lower LOQs of atractylenolides I, II, and III, and atractyloside A in rat plasma.

Compounds	Range (ng/mL)	Calibration Curves	Correlation Coefficient (*r*)	LLOQ (ng/mL)
Atractylenolide I	3.2–350	*y* = 0.00563*x* + 0.00578	*r* = 0.9919	3.2
Atractylenolide II	4–500	*y* = 0.00194*x* + 0.00347	*r* = 0.9927	4
Atractylenolide III	4–500	*y* = 0.00288*x* + 0.00839	*r* = 0.9977	4
Atractyloside A	3.44–430	*y* = 0.00165*x* + 0.00587	*r* = 0.9922	3.44

**Table 2 molecules-23-03234-t002:** Precision and accuracy of atractylenolides I, II, and III, and atractyloside A in rat plasma.

Compounds	Added C (ng/mL)	Intra-Day (*n* = 6)			Inter-Day (*n* = 18)		
		Measured C (ng/mL)	RSD (%)	RE (%)	Measured C (ng/mL)	RSD (%)	RE (%)
Atractylenolide I	5.6	5.62 ± 0.1	1.78	0.36	5.53 ± 0.15	2.71	1.25
	28	28.07 ± 1.36	4.85	0.25	27.83 ± 1.67	6	0.61
	140	141.53 ± 0.64	0.45	1.09	140.44 ± 0.82	0.58	0.31
Atractylenolide II	8	8.19 ± 0.08	0.98	2.38	8.09 ± 0.10	1.24	1.13
	40	40.43 ± 0.95	2.35	1.08	39.77 ± 1.16	2.92	0.56
	200	200.57 ± 1.12	0.56	0.29	199.32 ± 1.40	0.7	0.34
Atractylenolide III	8	8.22 ± 0.06	0.73	2.75	8.19 ± 0.1	1.22	2.38
	40	40.67 ± 0.59	2.46	1.68	40.43 ± 0.88	2.18	1.08
	200	200.53 ± 2.55	1.27	0.27	200.2 ± 3.96	1.98	0.1
Atractyloside A	6.88	6.48 ± 0.07	1.08	−5.81	6.09 ± 0.15	2.46	−5.81
	17.2	17.23 ± 1.26	7.31	−0.49	17.06 ± 1.57	9.2	−0.49
	172	172.63 ± 0.36	0.21	0.36	171.59 ± 0.60	0.35	0.36

**Table 3 molecules-23-03234-t003:** Mean extraction recovery and matrix effects of atractylenolides I, II, and III, atractyloside A, and acetaminophen in rat plasma.

Compounds	Added C (ng/mL)	Extraction Recovery (%)	RSD (%)	Matrix Effects (%)	RSD (%)
Atractylenolide I	5.6	87.81 ± 4.82	5.49	106.78 ± 0.85	0.80
	28	91.79 ± 1.17	1.27	90.09 ± 12.27	13.62
	140	82.04 ± 2.1	2.64	73.40 ± 1.07	1.45
Atractylenolide II	8	85.08 ± 5.30	6.23	99.48 ± 6.53	6.56
	40	96.57 ± 6.99	7.24	86.80 ± 11	12.67
	200	97.93 ± 1.90	1.94	84.96 ± 0.31	0.36
Atractylenolide III	8	85.76 ± 4.41	5.14	89.54 ± 9.79	10.93
	40	92.79 ± 2.04	2.19	92.69 ± 9.39	10.13
	200	80.26 ± 4.42	5.50	69.25 ± 4.72	6.82
Atractyloside A	6.88	82.47 ± 6.32	7.66	86.07 ± 6.49	7.54
	17.2	86.17 ± 2.41	2.80	115.30 ± 2.67	2.31
	172	71.90 ± 4.97	6.92	120.28 ± 7.29	6.06
4-Acetaminophen	52	98.54 ± 6.81	6.91	112.39 ± 4.33	3.85

**Table 4 molecules-23-03234-t004:** Stability of atractylenolides I, II, and III, and atractyloside A in rat plasma at room temperature.

Compounds	Added (ng/mL)	0 h Measured C (ng/mL)	4 h Measured C (ng/mL)	6 h Measured C (ng/mL)	RE (%)
Atractylenolide I	5.6	5.56	5.51	5.50	−1.37
	28	28.52	28.43	27.87	0.98
	140	140.49	140.45	139.61	0.13
Atractylenolide II	8	8.32	8.54	8.21	4.08
	40	40.88	40.06	40.49	1.19
	200	200.27	200.22	200.24	0.12
Atractylenolide III	8	8.06	8.02	7.96	0.17
	40	40.30	40.27	40.08	0.54
	200	200.47	200.31	199.05	−0.03
Atractyloside A	6.88	6.81	6.41	6.14	−6.2
	17.2	17.16	17.02	16.98	−0.85
	172	172.31	171.50	171.32	−0.17

**Table 5 molecules-23-03234-t005:** Freeze-thaw stability of atractylenolides I, II, and III, and atractyloside A in rat plasma.

Compounds	Added (ng/mL)	Zero Time Measured C (ng/mL)	One Time Measured C (ng/mL)	Two Time Measured C (ng/mL)	RE (%)
Atractylenolide I	5.6	5.95	5.04	5.02	−4.70
	28	28.42	28.50	28.01	1.11
	140	141.30	140.01	140.22	0.36
Atractylenolide II	8	8.78	8.46	8.28	6.33
	40	41.37	40.88	40.43	2.23
	200	201.26	200.04	200.63	0.32
Atractylenolide III	8	8.78	8.52	7.89	4.96
	40	40.10	40.48	40.63	1.01
	200	201.30	200.07	199.67	0.17
Atractyloside A	6.88	6.39	6.38	6.02	−8.96
	17.2	17.86	17.36	17.02	1.24
	172	172.73	171.48	171.65	−0.03

**Table 6 molecules-23-03234-t006:** AUC_0–∞_ and self-defined weighting coefficients of atractylenolides I, II, and III, and atractyloside A in raw and wheat bran-processed Atractylodis Rhizoma in rat plasma.

Compounds		Raw	Processed
Atractylenolide I	AUC_0–∞_ (μg·h/L)	130.44	230.74
	*W* _j_	0.18	0.22
Atractylenolide II	AUC_0–∞_ (μg·h/L)	202.37	218.88
	*W* _j_	0.28	0.21
Atractylenolide III	AUC_0–∞_ (μg·h/L)	233.09	289.03
	*W* _j_	0.32	0.27
Atractyloside A	AUC_0-∞_ (μg·h/L)	167.48	325.24
	*W* _j_	0.23	0.31

**Table 7 molecules-23-03234-t007:** Pharmacokinetic parameters of atractylenolides I, II, and III, and atractyloside A in rat plasma after oral administration of raw and wheat bran-processed Atractylodis Rhizoma.

Compounds	Groups	C_max_ (ng/mL)	T_max_ (h)	t_1/2_ (h)	AUC_0–t_ (h·μg/L)	AUC_0–∞_ (h·μg/L)	MRT_0–∞_ (h)
Atractylenolide I	Raw	32.09 ± 2.05	1.5 ± 0	3.58 ± 1.69	116.75 ± 18.38	130.44 ± 27.37	5.28 ± 1.36
	Wheat-bran processed	66.94 ± 10.89 *	1.5 ± 0	2.29 ± 1.18	219.14 ± 46.65 *	230.74 ± 44.79	4.86 ± 0.93 *
Atractylenolide II	Raw	49.62 ± 7.69	1.5 ± 0	4.12 ± 4.12	181.21 ± 29.35	202.37 ± 21.14	5.93 ± 3.14
	Wheat-bran processed	55.9 ± 13.58 *	1 ± 0	4.02 ± 3.10	202.43 ± 68.52 *	218.88 ± 61.97	5.03 ± 1.58
Atractylenolide III	Raw	87.04 ± 17.03	1 ± 0	1.56 ± 0.61	230.62 ± 76.76	233.09 ± 75.98	3.48 ± 0.29
	Wheat-bran processed	113.10 ± 19.04 *	1 ± 0	1.81 ± 0.79 *	284.83 ± 32.94 *	289.03 ± 32.52	3.07 ± 0.31 *
Atractyloside A	Raw	57.80 ± 21.65	0.95 ± 0.11	3.95 ± 2.22	138.41 ± 60.13	167.48 ± 90.09	6.05 ± 2.58
	Wheat-bran processed	69.38 ± 8.29 *	1.5 ± 0	2.55 ± 0.98	306.91 ± 73.75	325.24 ± 72.07	4.87 ± 0.84 *
Integrated data	Raw	54.17 ± 7.16	1 ± 0	2.473 ± 0.68	175.67 ± 28.09	184.54 ± 25.88	4.35 ± 0.33
	Wheat-bran processed	70.02 ± 5.16 *	1.5 ± 0	2.27 ± 0.36 *	263.23 ± 40.15 *	273.22 ± 39.99	4.14 ± 0.33 *

Atractylenolides I, II, and III, and atractyloside A in rats (mean ± SD, *n* = 5) after oral administration of raw and wheat bran-processed Atractylodis Rhizoma at a dose of 3.75 g/kg. * *p* < 0.05 vs. raw Atractylodis Rhizoma group.

**Table 8 molecules-23-03234-t008:** MRM parameters of atractylenolides I, II, and III, atractyloside A, and acetaminophen.

Compounds	Precursor Ion (*m*/*z*)	Product Ion (*m*/*z*)	Cone (V)	Collision (V)
Acetaminophen	151.9	110.16	42	14
Atractylenolide I	233.3	187.2	30	16
Atractylenolide II	231.2	185.2	34	14
Atractylenolide III	249.2	231.2	4	8
Atractyloside A	471.3	203.2	48	26

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
