# Peer review of "UPLC-MS/MS of Atractylenolide I, Atractylenolide II, Atractylenolide III, and Atractyloside A in Rat Plasma after Oral Administration of Raw and Wheat Bran-Processed Atractylodis Rhizoma"

_molecules, 2018, doi:10.3390/molecules23123234_

Reviewer 1 Report

The authors in this manuscript have determined the content of atractylenolidesI,II,III and atractyloside a in the plasma of rats that were orally administered. The manuscript is well designed, experiments well executed. Data is quite well presented. Definitely the manuscript would add value to the current  set of information. The authors  may need to come up with an answer to one question.

What is the rationale for performing the pharmacokinetic studies only up to 12 h. What happens to the state of the content of  atractylenolides at 24,48 and 72h. How are they metabolized. 

Also in the discussion the authors state wheat bran processing increases the spleen Tonifying activity of Atractylodis Rhizoma. Do the authors have any data to show or a reference to mention?

The discussion would look better if the pitfalls and future direction of the study are stated.

Author Response

Dear Reviewers:

Thank you for your comments on our manuscript entitled “Integral pharmacokinetic study and simultaneous determination by UPLC-MS/MS of atractylenolide I, atractylenolide II, atractylenolide III, and atractyloside A in rat plasma after oral administration of raw and wheat bran-processed Atractylodis Rhizoma” (Manuscript ID: molecules-397697). Those comments are all essential to improve the quality of our paper. We have carefully checked and revised our manuscript in accordance with your recommendations. On the hand, our response letter to you comments was finished by point to point way, which is listed following this letter.

 We are looking forward to hearing from you.

 With best wishes

Yours sincerely

Point 1: What is the rationale for performing the pharmacokinetic studies only up to 12 h. What happens to the state of the content of  atractylenolides at 24,48 and 72h. How are they metabolized. 

Response 1: Through literature review[1,2], we found that the T1/2 of  atractylenolides are mainly concentrated in 2-4 h, and it's almost completely metabolized within 12 hours. Furthermore, in the preliminary experiment, atractylenolides were difficult to be detected after 12 hours, and the results were not statistically significant.

[1] Zhu, Z.M.; Li, H.C.; Luo, J.B. Simultaneous determination of atractylenolide, , by HPLC-MS method and its application to pharmacokinetic study. Pharmacology and Clinics of Chinese Materia Medica. 2013, 29, 25-29.

[2] Jin, S.; Sun, X.H.; Gao, Z. Simultaneous determination of atractylenolide and in Rat Plasma by RP-HPLC. Strait Pharmaeeutieal Journal. 2010, 22, 36-38.

Point 2: Also in the discussion the authors state wheat bran processing increases the spleen Tonifying activity of Atractylodis Rhizoma. Do the authors have any data to show or a reference to mention?

Response 2: We cited the results of Xue's paper that in a dose-dependent manner, the bran-processed and crude AL increased the levels of TRY, AMS, VIP, and GAS and the mRNA expression levels of VIP. Compared with the crude AL, the processed Atractylodis Rhizoma is more effective in treating spleen deficiency syndrome  than the crude one. The reference had been inserted into the manuscript.

[1] Xue, D.H.; Liu, Y.Q.; Cai, Q.; Liang, K.; Zheng, B.Y.; Li, F.X.; Pang, X. Comparison of Bran-Processed and Crude Atractylodes Lancea Effects on Spleen Deficiency Syndrome in Rats. Pharmacogn Mag. 2018, 14, 214-219.

Point 3: The discussion would look better if the pitfalls and future direction of the study are stated.

Response 3: According to your opinion, the discussion of the manuscript had been added some new contents about the research pitfalls and future direction.

Reviewer 2 Report

In this manuscript, an UPLC-MS/MS method for simultaneous quantification of atractylenolides I, II, and III, and atractyloside A was established. This method was then successfully applied to compare these four active compounds in rat after oral administration of raw and wheat bran-processed Atractylodis Rhizoma. Comprehensive review has been provided in the introduction. The bioanalytical method optimization and validation details have been described. Given a similar study has been done by Zhu et al. with Atractylodis Macrocephalae Rhizoma, what is novel in this study? It’s unclear what are the differences between Atractylodis Macrocephalae Rhizoma and Atractylodis Rhizoma to readers. A few concerns please see below:

1.    Suggest move the optimized mass and HPLC parameters to methods.

2.    Figure 3- 7, please indicate the peak representing for the target analyte on the chromatograms, as there are multiple peak on the chromatographs. Otherwise, it’s very confusing for readers.

3.    The y-axis unit seems wrong. The intensity of background noise looks similar as the response of the spiked target (the max y values are all 100 for all chromatograms).

4.    Figure 3 is for atractylenolide I, the transition is 231.2>185.2 in Table I, while it becomes 233.3>187.2 on the Figure. The inconsistency issue is also observed between Table I and Figure 4.

5.    Table 2, the units for the linear range are lacking.

6.    Table 3, what does the “C” strand for?

7.    Method validation, how about the 4°C autosampler stability?

Author Response

Dear Reviewers: Thank you very much for your valuable comments on our paper “Integral pharmacokinetic study and simultaneous determination by UPLC-MS/MS of atractylenolide I, atractylenolide II, atractylenolide III, and atractyloside A in rat plasma after oral administration of raw and wheat bran-processed Atractylodis Rhizoma” (Manuscript ID: molecules-397697). We had revised the manuscript one by one in response to your comments. We hoped that our revision will enrich the content of the article and make the experimental research more meaningful. We are looking forward to hearing from you. With best wishes Yours sincerely

Point 1: Given a similar study has been done by Zhu et al. with Atractylodis Macrocephalae Rhizoma, what is novel in this study? It’s unclear what are the differences between Atractylodis Macrocephalae Rhizoma and Atractylodis Rhizoma to readers.

Response 1: The novelty of this study lies in the integrated pharmacokinetic analysis of four main compounds in Atractylodis Rhizoma. Both Atractylodis Macrocephalae Rhizoma and Atractylodis Rhizoma belong to the compositae plants, but in the theory of traditional Chinese medicine, the main efficacy of Atractylodis Macrocephalae Rhizoma is activating spleen, while Atractylodis Rhizoma is tonifying spleen. Shennong herbal classics record Atractylodis Rhizoma is mainly bitter, warm, dry and wet, applicable to damp turbidity internal resistance and partial to excess symptoms, there are sweating relief table, dispel wind and humidity and clear eyes, commonly used to treat cold with wet table syndrome, rheumatism bi syndrome, night blindness and eye dizziness and astringent syndrome. And Atractylodis Macrocephalae Rhizoma is mainly used to strengthen spleen and invigorate qi. It is suitable for those with spleen deficiency and dampness, but is partial to deficiency syndrome. It is also used for diuresis, antiperspirant and fetal placenta. [1]Chinese Pharmacopoeia Committee. Pharmacopeia of People’s Republic of China. Beijing, China: Chinese Medicine Science and Technology Publishing House. 2015. [2]Liu, J.;Jiang Y.G.; Hu, B.; Yong, X.J. Relevant Analysis of Pharmaceutical Syndrome of Baizhu Formulas. Journal of Chengdu Unversity of Tarditional Chinese Medicine. 2004, 04, 55-56.

Point 2: Suggest move the optimized mass and HPLC parameters to methods.

Response 2: We had moved the optimized mass and HPLC parameters to methods. 

Point 3: Figure 3- 7, please indicate the peak representing for the target analyte on the chromatograms, as there are multiple peak on the chromatographs. Otherwise, it’s very confusing for readers.

Response 3: Target peaks in Figure 3- 7 had been marked.

Point 4: The y-axis unit seems wrong. The intensity of background noise looks similar as the response of the spiked target (the max y values are all 100 for all chromatograms).

Response 4: The vertical coordinates in the figure are all displayed at 100%. The peaks with similar response values are solvent peaks, which decreases with the increase the content of standard substance. Some graphs are similar to each other in peak strength due to proportional adjustment. The chromatogram with the standard substance was detected with a peak at the corresponding retention time, while the blank plasma was not. In addition, due to the low content of components, the signal intensity of some detection graphs is too low, but it can still be identified after amplification. The results are reliable.

Point 5: Figure 3 is for atractylenolide I, the transition is 231.2>185.2 in Table I, while it becomes 233.3>187.2 on the Figure. The inconsistency issue is also observed between Table I and Figure 4.

Response 5: This is a clerical error and had been revised in the manuscript.

Point 6: Table 2, the units for the linear range are lacking.

Response 6: The unit of linear range is ng/mL and had been added to the manuscript.

Point 7: Table 3, what does the “C” strand for?

Response 7: The “C” stands for quality control sample

Point 8: Method validation, how about the 4℃ autosampler stability?

Response 8: When the temperature of 4℃, the repeatability (RSD value) of injection results was less than 3‰. Therefore, the autosampler has a good repeatability and high stability at this temperature. In addition, the samples have the best solubility at this temperature and the measurement result is accurate.

Round  2

Reviewer 2 Report

Comments are well addressed.